# Conceptualizing Vulnerability for Health Effects of the COVID-19 Pandemic and the Associated Measures in Utrecht and Zeist: A Concept Map

**DOI:** 10.3390/ijerph182212163

**Published:** 2021-11-19

**Authors:** Lilian G. L. van der Ven, Elisa L. Duinhof, Michel L. A. Dückers, Marielle Jambroes, Marja J. H. van Bon-Martens

**Affiliations:** 1Julius Center for Health Sciences and Primary Care, University Medical Center Utrecht, Utrecht University, 3584 CS Utrecht, The Netherlands; m.jambroes@umcutrecht.nl; 2Trimbos Institute (Netherlands Institute of Mental Health and Addiction), Department Mental Health and Prevention, 3521 VS Utrecht, The Netherlands; eduinhof@trimbos.nl (E.L.D.); mbon@trimbos.nl (M.J.H.v.B.-M.); 3Netherlands Institute of Health Services Research (NIVEL), 3513 CR Utrecht, The Netherlands; m.duckers@nivel.nl; 4Faculty of Behavioural and Social Sciences, University of Groningen, 9712 CP Groningen, The Netherlands

**Keywords:** COVID-19, vulnerability, concept mapping, community engagement

## Abstract

The COVID-19 pandemic and the associated measures have impacted the health of many. Not all population groups are equally vulnerable to such health effects, possibly increasing health inequalities. We performed a group concept mapping procedure to define a common, context-specific understanding of what makes people vulnerable to health effects of the pandemic and the measures. We organized a two-step, blended brainstorming session with locally involved community members, using the brainstorm focus prompt ‘What I think makes people vulnerable for the COVID-19 pandemic and the measures is…’. We asked participants to generate as many statements as possible. Participants then individually structured (sorted and ranked) these statements. The structuring data was analysed using the groupwisdom^TM^ software and then interpreted by the researchers to generate the concept map. Ninety-eight statements were generated by 19 participants. Sixteen participants completed both structuring tasks. The final concept map consisted of 12 clusters of vulnerability factors, indicating a broad conceptualization of vulnerability during the pandemic. It is being used as a basis for future research and local supportive interventions. Concept mapping is an effective method to arrive at a vulnerability assessment in a community in a short time and, moreover, a method that promotes community engagement.

## 1. Introduction

The COVID-19 pandemic and the associated measures impact and will likely continue to impact the physical, mental and social health of many [1,2,3]. In the Netherlands, there is an estimated loss of 34,000 to 50,000 healthy life years due to postponed treatments and an estimated 200–700 extra deaths from smoking [4,5], mental health was at a low point in the first half of 2021 [6] and mental well-being decreased with stricter measures, while loneliness increased [7].

Not everyone is equally vulnerable to such health effects. When it comes to the disease itself, we know that the elderly and people with underlying conditions are more at risk for a severe course of COVID-19 and death [8,9], while people with a migration background have a two times higher risk of being infected with the virus than people without a migration background [10]. The impact of the pandemic and the measures can also be particularly severe for those who are most vulnerable [11,12]; they are at risk for the social and health consequences, both direct and indirect, of the COVID-19 crisis approach [13]. The pandemic interacts with existing inequalities in the social determinants of health and in the prevalence of chronic diseases, making this pandemic a syndemic [14]. As a result, existing health inequalities could increase even further [15].

This syndemic is not universal; the way existing vulnerabilities are expressed and interact with the pandemic is context-dependent [16]. Vulnerability can be seen as a multifactorial and context-dependent concept, in which people can have one or more layers of vulnerability [17]. So, if we want to examine the health impact of the COVID-19 pandemic for vulnerable groups in a certain community—a neighbourhood, a city, a country—a necessary first step is to map what vulnerabilities mean locally for the population during the pandemic. This mapping should be done by the community members themselves—citizens, the professionals who support them, intermediaries, policy makers, and researchers—as it is a form of community engagement that acknowledges the value of local knowledge and promotes the feeling of ownership in the community, ‘nothing about me without me’ [18]. Community engagement has been shown to play a central role in controlling outbreaks and other public health emergencies [19,20], and to promote support for future interventions [21]. In this way, methods from the social sciences can be used to identify vulnerabilities within communities during a pandemic [22].

A fast and transparent method to accomplish this is concept mapping. Concept mapping is a participatory research method that makes optimal use of different knowledge sources [23]. It is a structured procedure that aims to examine and integrate the knowledge and ideas of groups of people on a complex topic. By engaging different stakeholders in an early stage, it increases the chance of successful implementation of future interventions that build on the concept map [24]. It has previously shown to be an effective and efficient strategy to clarify complex mental health topics [25] and to be of value for evidence based public health policy by integrating practical knowledge with scientific knowledge [26]. It also helps to map out connections between different aspects of a concept, in order to get a fine-grained perspective on the different facets of the concept. A pooled analysis of 69 studies concludes that concept mapping yields strong internal representational validity and very strong sorting and ranking reliability estimates [27].

In this paper, we describe a concept mapping procedure to define a common understanding of what makes people vulnerable to the health effects of the COVID-19 pandemic and the associated measures, thereby integrating the knowledge and views of citizens, professionals and researchers in the cities of Utrecht and Zeist in the Netherlands. The aim of this study was to arrive at a context-dependent concept map of vulnerability within this community.

## 2. Materials and Methods

In this study, we followed the six-step procedure of Trochim [23]: (1) preparation, (2) generation of statements, (3) structuring of statements, (4) graphical representation of statements, (5) interpretation, and (6) implementation. We used the groupwisdom^TM^ software [28] to perform the procedure.

### 2.1. Step 1: Preparation

The study was performed in the cities of Utrecht and Zeist, two neighbouring cities in the province of Utrecht, the Netherlands. Utrecht is the fourth largest city in the Netherlands, with just over 300,000 inhabitants, a third of whom have a migration background. Its population is relatively young, with 64,000 university students. Zeist is a smaller city with around 54,000 inhabitants, about a quarter of whom have a migration background [29,30]. It is located in the green, wooded areas to the east of Utrecht. Figure 1 shows where Utrecht and Zeist are located in the Netherlands.

We chose to focus on Utrecht and Zeist because of the aforementioned importance of context when defining vulnerability. These cities represent variation—large and urban versus small and more rural—within a similar context: they are adjacent and located in the same region, and they both have neighbourhoods with higher and lower socioeconomic status. Both Utrecht and Zeist, like the rest of the Netherlands were confronted with waves of infections in the first half of 2020 and measures, such as social distancing, hygiene measures and lockdowns. Furthermore, in these two municipalities, science, practice and policy have been working together for some time to jointly tackle socially relevant issues from multiple knowledge sources. Local policy makers and professionals wanted to work on mitigating the health impact of the pandemic, and concept mapping was seen as a way to formulate a common basis for planning and evaluation, and thus to promote support for future interventions.

Because of the COVID-19 measures that were in place, we decided to perform the procedure online. We used purposive sampling, focusing on recruiting participants with as many different perspectives and backgrounds as possible from the broad network of the researchers. We invited participants representing three groups: citizens, professionals and researchers. By citizens, we mean people living in the cities of Utrecht and Zeist who belong to groups that are typically recognized as being vulnerable for health inequalities (e.g., people with low socioeconomic status, low literacy, mental health problems or a chronic illness, and migrants), and groups of people that might have become vulnerable for health loss due to the COVID-19 pandemic and the associated measures (e.g., people with loss of income, job loss, or loss of social contacts). By professionals, we mean professionals working for organisations that focus on supporting various vulnerable groups in Utrecht and Zeist.

### 2.2. Step 2: Generation of Statements

In November 2020, we organised a two-step concept mapping brainstorming session. To ensure that traditional vulnerable groups were not excluded or underrepresented as the session might be too long or difficult for them, we used a two-step, blended approach: two professionals collected statements during a live brainstorming session with vulnerable citizens and brought these statements in during the online session. Furthermore, to ensure that hosting the session online was not a barrier for participants to join, we developed a manual with instructions on how to log in to the session, solutions for common technical problems and a telephone number participants could call for help.

During the online session, after the introduction, participants were asked to turn off their cameras for 10 min and write down their answers to the brainstorm focus prompt ‘From my perspective, what makes people vulnerable for the COVID-19 pandemic and the accompanying measures is …’. After this individual reflection, we asked participants to share their two most important answers in turn. A scribe noted these statements on a shared screen. Overlapping statements were deleted, and statements were checked for clarity, unambiguity, singularity and equality in consultation with the participants. Next, participants were asked to bring up remaining statements that they did not yet hear from others during the session. The session was ended with a short presentation on the next steps of the concept mapping procedure. The session was recorded to enable checking for accuracy and editing of the noted statements.

After the session, we collected the statements that were typed in the chat during the session. Participants could also e-mail any additional statements that were not discussed during the brainstorm. In addition, we invited participants that could not join the session to e-mail their statements to the researchers, and if these statements were new, they were added to the statements collected during the session. All added statements were also checked for clarity, unambiguity, singularity and equality.

### 2.3. Step 3: Structuring of Statements

For the structuring of the statements, we invited persons that were present in the online brainstorming session, persons that e-mailed their statements afterward and persons that were invited to the online session but did not attend nor e-mail their statements. Participants were invited to perform two structuring tasks: sorting and ranking. First, participants were asked to sort the statements into groups based on similar meaning or content and give each of these groups a description. All statements needed to be sorted, and statements could only be sorted once. Next, participants were asked to rank each statement on its importance on a scale from one (least important) to five (most important). Participants could complete these tasks online individually, but we organised an online help session for those who needed it. Participants who could not join the brainstorming session were also invited to perform the structuring tasks.

To assure anonymity, random usernames and passwords were generated for all participants, and personal information and anonymous research data were stored separately. Only one researcher, who was not involved in analysing the data, had insight into which username and password belonged to which participant. After completing the structuring of the statements, participants that partook in both the brainstorming session and the structuring received a gift card of EUR 30.

### 2.4. Step 4: Graphical Representation of Statements

After the structuring of the statements was finished, we analysed the data using the groupwisdom^TM^ software (Concept Systems, Inc., Ithaca, NY, USA) using multidimensional scaling and hierarchical cluster analyses. First, a point map was made, in which each statement is reflected by a point. The more often the participants grouped two statements together in the same pile, the smaller the distance between two points on the map and so the more similar the statements are in terms of content or meaning. Second, a point rating map was produced, in which the size of each point indicates the perceived importance of the statement. Third, a cluster map was created, which reflects the sorting of the statements. The software produces several cluster solutions and suggestions to merge clusters in order to reduce the number of clusters.

### 2.5. Step 5: Interpretation

The cluster map solutions, cluster merge information, and cluster label suggestions of the participants were discussed by the five researchers involved in the project. Starting from one of the suggested cluster solutions and following the order of the cluster merge info (cluster tree) produced by the groupwisdom^TM^ software, we extensively discussed each suggested merge, assessing whether these clusters should be merged based on similar meaning of the statements without losing meaningful distinctions. This resulted in a final cluster solution. Second, the cluster labels of the remaining clusters were extensively reviewed and discussed. We occasionally consulted the spanning info (information how frequently a statement has been sorted with any other statement) and bridging values (indicating if the statement was more often sorted with statements that were close or more distant on the map), to see if some statements should be moved to another cluster. Finally, a cluster rating map was produced, reflecting the average ranking of the clusters.

### 2.6. Step 6: Implementation

We reflect upon the implementation and utilization of our concept map in the Discussion chapter of this paper.

## 3. Results

### 3.1. Participants

A flowchart of the participants in the various steps of the procedure is shown in Figure 2. First, two professionals collected statements from five citizens during a live brainstorming session. These professionals then joined the online brainstorming session. In this online session, a total of thirteen participants representing the perspectives of citizens, professionals and researchers participated. Six participants that could not join the session e-mailed their statements afterward.

Out of the nineteen participants that were invited to complete the structuring tasks, fourteen completed both tasks and five did not. Six participants did not participate in the brainstorming session, but did complete the structuring tasks. This resulted in a total number of sixteen participants that rated and sorted the statements: six representing the perspective of citizens, seven representing the perspective of professionals and three representing the perspective of researchers.

### 3.2. Statements and Clusters

The statements generated during the brainstorming session and the statements e-mailed afterward added up to a total of 196. Removal of duplicates reduced this to 98 statements that were used for sorting and ranking. For the interpretation of the results, we started with a 15-cluster solution that was suggested by the groupwisdom^TM^ software. Subsequent merging led to a final 12-cluster solution. Consulting of the spanning info and bridging values did not lead to changes to the clusters. Table 1 shows the ranking of each cluster. Appendix A shows the statements in each cluster and the average ranking of the statements.

### 3.3. Description of the Concept Map

The final concept map is shown in Figure 3. The statements or vulnerability factors are plotted along two axes: from endogenous factors (e.g., knowledge about and attitude towards COVID-19) to exogenous factors (e.g., not being able to do volunteer work anymore), and from current impact (e.g., the lack of informal help) to impact in time (e.g., losing your job). The stress value for the multidimensional scaling is 0.3186.

The cluster that participants find most important is ‘consequences for health and care’ (e.g., avoiding or delaying care), with an average ranking of 4.21 out of 5. Other high-ranking clusters are ‘personal environment’ (3.83, e.g., being in an unsafe (home) situation), ‘finances’ (3.70, e.g., having financial worries) and ‘work and income’ (3.68, e.g., losing your job). ‘Perception of work’ (e.g., prolonged working from home) is considered the least important cluster, with an average rating of 2.79.

### 3.4. Comparing the Perspectives

In Figure 4, we compared the average ranking of the clusters for the three participant perspectives: citizens, professionals and researchers. The figure shows how each perspective ranked the clusters relative to each other. For citizens and professionals, ‘consequences for health and care’ was the most important cluster. For researchers, ‘finances’ was the most important. For all perspectives, the least important cluster was ‘perception of work’. Overall, citizens and researchers had fairly similar rankings while professionals differed, attaching less importance to for example ‘finances’, ‘mental health’ and ‘work and income’, and more to ‘social environment’ and ‘activities’.

## 4. Discussion

In this study, through the process of concept mapping, locally involved community members arrived at a definition of vulnerability in which different themes can be distinguished, and a number of priorities can be identified. Our concept map indicates a broad conceptualization of vulnerability for health effects of the COVID-19 pandemic, with clusters of vulnerability factors being plotted along two axes: from endogenous to exogenous factors and from current impact to long-term impact. The elements of vulnerability in this concept map reflect the social determinants of health that can traditionally be distinguished in public health literature [32,33] and match definitions of vulnerabilities in the face of other threats, such as disasters and heat waves [34,35]. Moreover, they are in line with recent studies that aimed to shed light on health vulnerabilities highlighted by the COVID-19 pandemic as well. Several authors so far have shown how the pandemic disproportionately affects the health of poor, marginalized ethic groups, women, people working from home or being unemployed and those with already poor health [36,37,38,39,40]. We also show the interrelationship between these vulnerability factors by placing them in a two-dimensional surface.

When comparing the perspectives, professionals rank the clusters differently from citizens and researchers, placing more emphasis on social environment and activities and less on finances, mental health and work and income. A similar difference between healthcare providers (specifically physicians) and patients was found in a study on the perceived importance of various factors for the concept of health [41]. It is important to keep these differing priorities in mind when thinking about interventions to support vulnerable groups; interventions should align with their priorities.

### 4.1. Implementation

We show that it is possible to arrive at a common, supported conceptualization of vulnerability, as a basis from which we can start working on solutions to support vulnerable populations. The challenge is to continue to use the same community engagement and empowerment philosophy so that the identified vulnerabilities can be reduced [42]. We have used our concept map to identify indicators of the 12 clusters of vulnerability. Based on these indicators, we have selected three neighbourhoods in Utrecht and Zeist that differ in the prevalence of these indicators and thus represent different vulnerability factors. In these neighbourhoods, we have used the concept map as a base for dialogue with professionals about which groups of citizens they think are the most vulnerable in their neighbourhood, how to identify and reach these citizens and how to involve them in finding solutions. We are now interviewing these citizens about their vulnerabilities, the effects of the pandemic on their health and their health and care needs, using the concept map as input for the topics we discuss in these interviews.

In this way, our concept map helps us to gain knowledge about the health impact of the pandemic for vulnerable groups, both by identifying vulnerable groups and by giving direction to the conversation about how the pandemic and the associated measures affect their health. We want to translate this knowledge into solutions. There already are various initiatives at the local level to support vulnerable groups and thus mitigate the negative impact of the pandemic. We will use the knowledge we have gained in this project to identify, strengthen and improve promising strategies for local prevention, care and support. We will do this in co-creation with citizens, professionals and policy-makers in the community.

This is especially important for the next step. The potential of this type of vulnerability assessment work for actual risk reduction depends on the degree to which governments, public and private organizations and other actors in local communities—and at different levels where conditions are influenced—manage to resolve vulnerabilities that can be linked to causes that can be embedded in complex and structural problems. Although this is something that falls outside the scope of our study, we do recommend that assessments like these are followed up deliberately and that the quality of problem-solving efforts as well as enablers and barriers in this process are evaluated.

### 4.2. Strengths and Limitations

A strength of the concept mapping method used in this study remains the involvement and dialogue between those involved, and hence the compatibility with community engagement—or formulated differently: assessing vulnerabilities based on the concept mapping approach is an example of a social science-driven structured community engagement strategy [22]. A second strength is that we found a creative way to deal with the risk of excluding or underrepresenting those with limited digital skills by using a two-step, blended approach with a live brainstorming session with vulnerable citizens before the online brainstorming session.

Online group brainstorming does have limitations. It is harder to facilitate discussions among participants in an online environment than it would be in a live session. This could have been counteracted by letting participants perform the brainstorming individually, for example by letting them add and react individually to statements generated by other participants. However, the added value of group brainstorming is that there is room for clarifying questions, and participants can together ensure clarity, unambiguity, singularity and equality of the statements, which in turn increases the chance that the structuring takes place based on the unambiguous meaning of the statements. This is reflected in the stress value for the multidimensional scaling—indicating the statistical fit of the two-dimensional map, given the structuring data of participants—of 0.3186, which falls within the range of stress values that 95% of concept mapping procedures yield (0.205–0.365) [43].

Based on our results, we do find that concept mapping is suitable to be used online if necessary, but with fewer participants than in a live session. This means we could include a smaller number of participants than we would have included in a live session. We counteracted this by having a separate live brainstorming session which generated statements from five extra citizens, and by giving participants the chance to contribute statements via e-mail even if they had not participated in the online session. In this way, we have met the methodological recommendations of having at least ten participants with a broad representation of the relevant perspectives for the issue at hand [43].

A second limitation is that our study uses a purposive sample, which can be prone to selection bias. We tried to eliminate this to the best of our ability by extensively discussing the possible participants, focusing on identifying participants with as many different backgrounds and relevant perspectives as possible.

Thirdly, because vulnerability is a context-dependent concept, we do not know the extent to which the concept map we created in our community is applicable to other contexts. It was not the aim of our study to create a broadly generalizable concept map. As mentioned above, the clusters of vulnerability factors we found are in line with other studies about vulnerabilities during the COVID-19 pandemic, which makes it plausible that our results have validity in other contexts. We involved a variety of stakeholders and perspectives, which also strengthens the external validity [24]. However, it would be interesting to repeat this procedure in other communities to find the similarities and differences.

Lastly, as social contexts, especially during crises, are changeable, it is important to continue to verify outcomes and see if they are still applicable over time.

### 4.3. Future Perspectives

We will use our concept map to design follow-up research on the health effects of the pandemic in vulnerable groups, and to guide decision making on the local level for supportive interventions. In all of these activities, community engagement remains a key priority.

Apart from the value it has for further risk reduction activities in the locality of the Utrecht region, our study also shows that concept mapping is an effective and cost-effective method to arrive at a vulnerability assessment in a community in a short time. This complements the globally growing toolbox of tools, such as Napier’s barefoot approach [44] and other tools as described by Jeleff and others [45]. The method has proved useful in the context of a Dutch community during the COVID-19 pandemic, but it would be interesting to see to what extent it leads to comparable insights into vulnerabilities in other communities and countries during COVID-19 or other infectious diseases, or other disasters and calamities.

## 5. Conclusions

In this study, together with locally involved community members in Utrecht and Zeist we generated a concept map of vulnerability, which indicates a broad conceptualization of vulnerability for the health effects of the COVID-19 pandemic and the associated measures. Our concept map is used as a common, supported basis for future research and interventions. We found that concept mapping is an effective method to arrive at a vulnerability assessment in a community in a short time, and moreover, a method that promotes community engagement. As applies to any vulnerability assessment, a crucial next step is to ensure that communities are assisted in an adequate response and that lessons are drawn on behalf of future community-based vulnerability reduction strategies.

## Figures and Tables

**Figure 1 ijerph-18-12163-f001:**
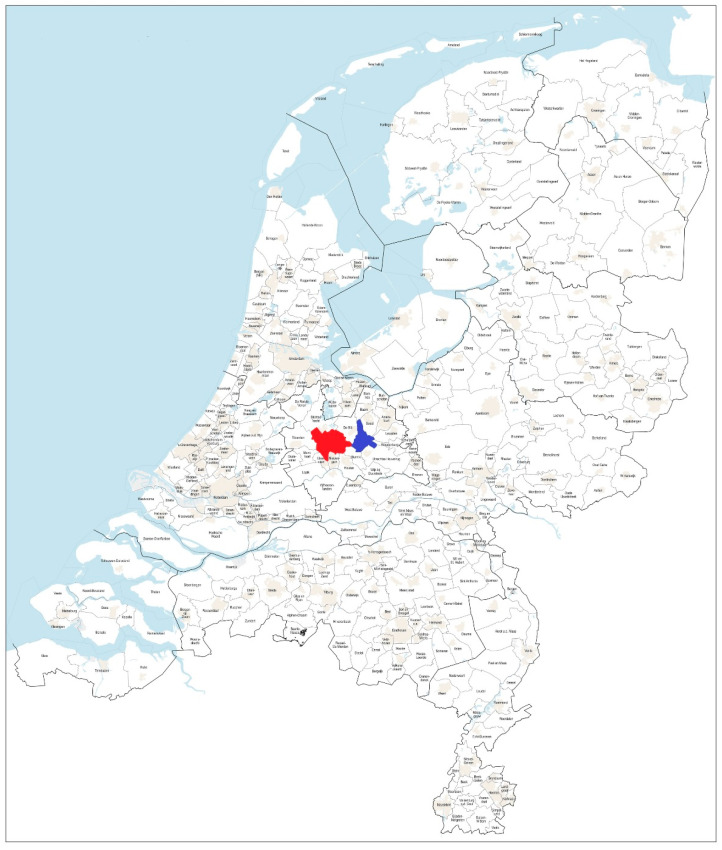
Map of the Netherlands with Utrecht shown in red and Zeist shown in blue. Based on a basic map showing the 352 Dutch municipalities as of 1 January 2021 [31].

**Figure 2 ijerph-18-12163-f002:**
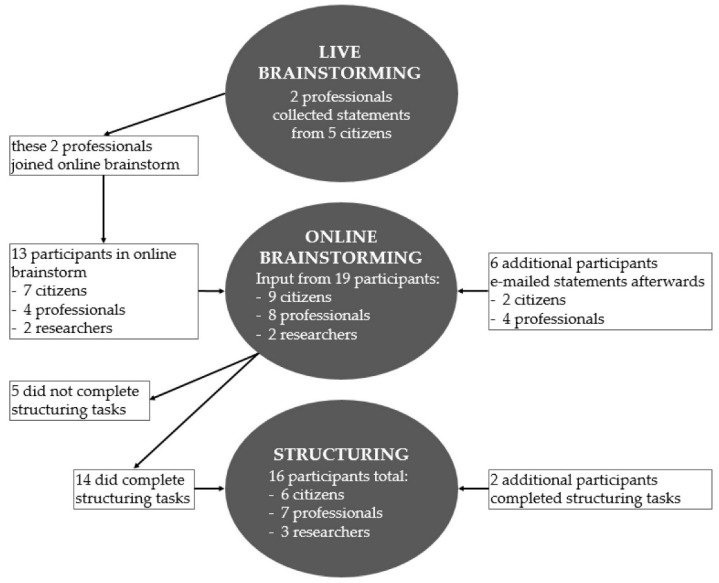
Flowchart of participants in brainstorming sessions and structuring tasks.

**Figure 3 ijerph-18-12163-f003:**
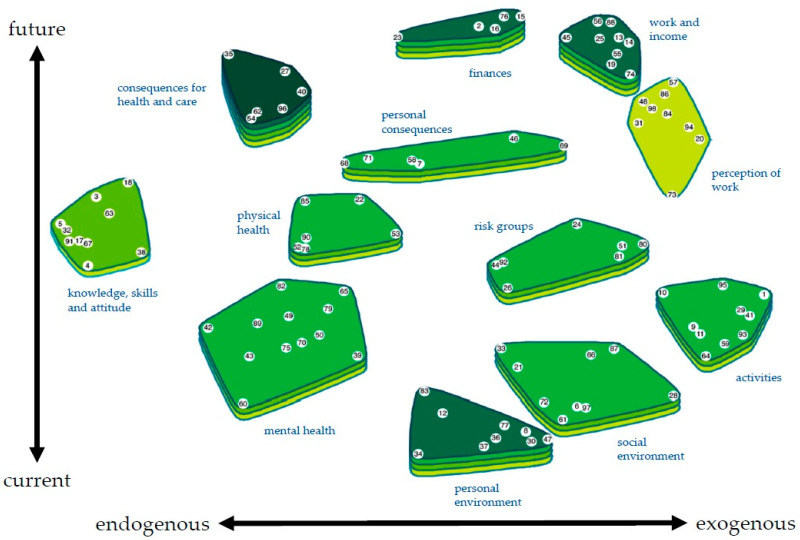
Final concept map with 12 clusters plotted along two axes. A darker green colour indicates a higher average ranking of the cluster. Each numbered point indicates a statement.

**Figure 4 ijerph-18-12163-f004:**
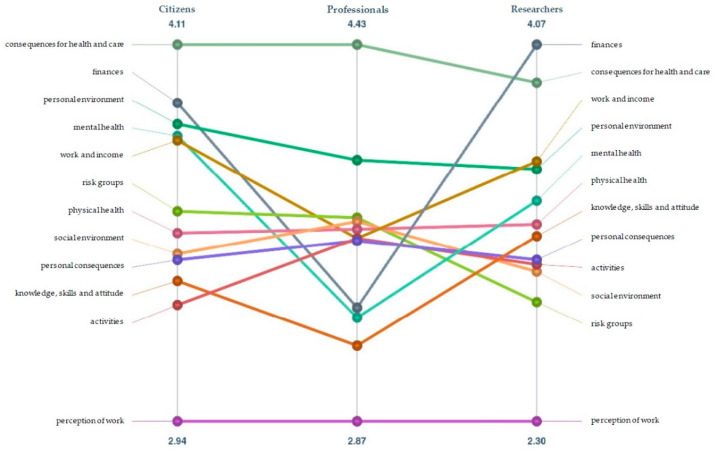
Comparison of average ranking of clusters for citizens, professionals and researchers.

**Table 1 ijerph-18-12163-t001:** Clusters in final 12-cluster solution and average ranking of clusters.

No.	Cluster Label	Average Ranking ^1^
1	Activities	3.40
2	Risk groups	3.51
3	Social environment	3.48
4	Personal environment	3.83
5	Finances	3.70
6	Work and income	3.68
7	Perception of work	2.79
8	Knowledge, skills and attitude	3.25
9	Mental health	3.50
10	Personal consequences	3.45
11	Physical health	3.53
12	Consequences for health and care	4.21

^1^ On a scale from 1.00–5.00.

## Data Availability

All of the available data is included in the article and the Appendix A.

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
