# Peer review of "Conceptualizing Vulnerability for Health Effects of the COVID-19 Pandemic and the Associated Measures in Utrecht and Zeist: A Concept Map"

_ijerph, 2021, doi:10.3390/ijerph182212163_

Round 1

Reviewer 1 Report

The reviewed article addresses the current issue of COVID-19. It is generally written correctly, its structure is appropriate. It presents the application of one of the methods of social research on the example of two cities. However, I have a few comments that I think should be taken into account for the article to be published.

  1. The title does not indicate the use of a case study of two neighbouring cities Utrecht and Zeist in the Netherlands. One can debate whether the case study is too narrow to draw broader conclusions. Of course, one may be interested in this work because of the subject matter undertaken and the method used. However, it would have been better to expand the research area, e.g. to cities with different local conditions and make a comparison, or to refer to the case study in the title.
  2. The introduction should contain more references to the literature. After all, a great deal of work has been written on COVID-19. Perhaps it would be a good idea to collate them problematically. Besides, it would be worthwhile to provide more references to the applications of the concept map method. Here, too, the aim of the article should be clearly defined.
  3. In chapter 2, Materials and methods, it is written that the study was carried out in the cities of Utrecht and Zeist. Unfortunately, there is no more information about these cities in the article - where are they located, what is their size, what are the local conditions, what is their functional-spatial structure, what are the social and other problems, how is the COVID-19 pandemic going on here? Are these cities representative of other cities? In my opinion, it is necessary to justify the choice of cities. After all, the introduction points out the importance of context, or just social context? Maybe it is also worth adding a figure with the location of the cities?
  4. The small number of participants in the study (especially citizens) needs to be justified. Is it enough to draw general conclusions, even if only concerning the application of the method?
  5. The conclusions chapter consists of only two sentences. This is definitely too little. Maybe it would be better to combine it with the preceding chapter Discussion and expand it?

I encourage you to make corrections and continue your research.

Reviewer 2 Report

The authors aim at defining a common understanding of what makes people vulnerable to the health effects of the COVID-19 pandemic; to reach this goal, they use qualitative methods, above all different ways and levels of brainstorming. The paper is quite topical and surely interesting for readers; however, the results of the study are unsurprising and rather unsophisticated.

I have some concrete questions and concerns:

  • Figure 1 seems not to be in agreement with the description above it. The problem is the block in the upper right corner. Those 6 additional participants are not added to 13 participants of the basic brainstorming. Moreover, 6 additional participants are composed of 7 citizens and 4 professionals, which is in contradiction with basic arithmetics.
  • The Discussion section is not satisfactory. It contains some basic discussion on the methods. However, there is no discussion on the impact of the results on COVID-19 management. What is the value of the results the authors reached, and how can be implemented (compare point 6 on line 66 - there is no word about implementation neither in Results nor in Discussion). The results of the paper should be also compared with the results published earlier.

Round 2

Reviewer 2 Report

The authors improved the paper substantially, above all by including different pieces of background information. Moreover, they corrected the errors I pointed at. As concerns the newly enlarged discussion, now it is satisfactory. In the present form, the paper is acceptable for publishing.